

**N$_2$O$_5$ uptake onto saline mineral dust: a potential missing source of tropospheric**
**ClNO$_2$ in inland China**
Haichao Wang[1,#], Chao Peng[2,#], Xuan Wang[3,#], Shengrong Lou[4], Keding Lu[5], Guicheng Gan[6],
Xiaohong Jia[1], Xiaorui Chen[5], Jun Chen[6], Hongli Wang[4], Shaojia Fan[1,7], Xinming Wang,[2,8,9]
Mingjin Tang[2,8,9,*]
[1] School of Atmospheric Sciences, Sun Yat-sen University, Guangzhou, China
[2] State Key Laboratory of Organic Geochemistry, Guangdong Key Laboratory of Environmental
Protection and Resources Utilization, and Guangdong-Hong Kong-Macao Joint Laboratory for
Environmental Pollution and Control, Guangzhou Institute of Geochemistry, Chinese Academy of
Sciences, Guangzhou, China
[3] School of Energy and Environment, City University of Hong Kong, Hong Kong SAR, China
[4] State Environmental Protection Key Laboratory of Formation and Prevention of the Urban Air
Complex, Shanghai Academy of Environmental Sciences, Shanghai, China
[5] State Key Joint Laboratory of Environmental Simulation and Pollution Control, College of
Environmental Sciences and Engineering, Peking University, Beijing, China
[6] Institute of Particle and Two-Phase Flow Measurement, College of Energy and Power
Engineering, University of Shanghai for Science and Technology, Shanghai, China
[7] Guangdong Provincial Observation and Research Station for Climate Environment and Air
Quality Change in the Pearl River Estuary, Key Laboratory of Tropical Atmosphere-Ocean System,
Ministry of Education, Southern Marine Science and Engineering Guangdong Laboratory
(Zhuhai), Zhuhai, China



[8] CAS Center for Excellence in Deep Earth Science, Guangzhou 510640, China
[9] University of Chinese Academy of Sciences, Beijing, China
[#]These authors contributed equally to this work
*correspondence: Mingjin Tang (mingjintang@gig.ac.cn)





**Abstract**
Nitryl chloride ($ClNO_2$), an important precursor of Cl atoms, significantly affects atmospheric
oxidation capacity and $O_3$ formation. However, sources of $ClNO_2$ in inland China have not been
fully elucidated. In this work, laboratory experiments were conducted to investigate heterogeneous
reaction of $N_2O_5$ with eight saline mineral dust samples collected from different regions in China,
and substantial formation of $ClNO_2$ was observed. $ClNO_2$ yields, $\varphi(ClNO_2)$, showed large
variations (ranging from <0.05 to ~0.77) for different saline mineral dust samples, largely
depending on mass fractions of particulate chloride. In addition, for different saline mineral dust
samples, $\varphi(ClNO_2)$ could increase, decrease or show insignificant change as RH increased from
18% to 75%. We further found that current parameterizations significantly overestimated $\varphi(ClNO_2)$
for heterogeneous uptake of $N_2O_5$ onto saline mineral dust. Assuming a uniform $\varphi(ClNO_2)$ value
of 0.10 for $N_2O_5$ uptake onto mineral dust, we used a 3-D chemical transport model to assess the
impact of this reaction on tropospheric $ClNO_2$ in China, and found that weekly mean nighttime
maximum $ClNO_2$ mixing ratios could be increased by up to 85 pptv during a severe dust event in
May 2017. Overall, our work showed that heterogeneous reaction of $N_2O_5$ with saline mineral dust
could be an important source of tropospheric $ClNO_2$ in inland China.



## 1 Introduction

The formation of $O_3$ and secondary aerosols, two major air pollutants, is closely related to atmospheric oxidation processes (Lu et al., 2019). Primary pollutants emitted by natural and anthropogenic sources are oxidized by various oxidants to produce $O_3$ and secondary aerosols, affecting air quality and climate. Major tropospheric oxidants include OH radicals, $NO_3$ radicals and $O_3$, and in the last two decades Cl atoms have been proposed as an important oxidant (Saiz-Lopez and von Glasow, 2012; Simpson et al., 2015; Wang et al., 2019). Rate constants for reactions of certain volatile organic compounds (VOCs) with Cl atoms can be a few orders of magnitude larger than those reacting with OH radicals (Atkinson and Arey, 2003; Atkinson et al., 2006); therefore, despite its lower concentrations in the troposphere, Cl can contribute significantly to atmospheric oxidation capacity in some regions (Saiz-Lopez and von Glasow, 2012; Simpson et al., 2015; Wang et al., 2019). For example, a modeling study (Sarwar et al., 2014) suggested that including Cl chemistry in the model could enhance oxidative degradation of VOCs by >20% in some locations.

One major source of tropospheric Cl atoms is daytime photolysis of $ClNO_2$ (Thornton et al., 2010; Simpson et al., 2015), which is formed in heterogeneous reaction of $N_2O_5$ with chlorine-containing particles (R1) at nighttime (Osthoff et al., 2008; Thornton et al., 2010):

$$N_2O_5(g) + Cl^-(aq) \rightarrow \varphi ClNO_2(g) + (2-\varphi)NO_3^-(aq) \qquad (R1)$$

The uptake coefficient, $\gamma(N_2O_5)$, and the $ClNO_2$ yield, $\varphi(ClNO_2)$, both depend on relative humidity (RH), aerosol composition and mixing state, and etc. (Bertram and Thornton, 2009; Ryder et al., 2014; Mitroo et al., 2019; McNamara et al., 2020; Yu et al., 2020). Cl atoms produced by $ClNO_2$ photolysis can effectively enhance atmospheric oxidation (Le Breton et al., 2018; Wang et al., 2019) and thus increase concentrations of $O_3$ and OH radicals during the day (Simon et al., 2009;



Riedel et al., 2014; Sarwar et al., 2014; Tham et al., 2016; Wang et al., 2016). In addition, ClNO$_2$
is an important temporary reservoir of NO$_x$ at night and releases NO$_2$ during the daytime via
photolysis, thereby further affecting daytime photochemistry.

Sea spray aerosol is the most important source of particulate chloride (Cl$^-$), and ClNO$_2$ is

expected to be abundant at marine and coastal regions impacted by anthropogenic emissions. High
levels of ClNO$_2$ have been observed at various marine and coastal regions over the globe (Simon
et al., 2009; Riedel et al., 2012; Tham et al., 2014; Young et al., 2014; Wang et al., 2016; Osthoff
et al., 2018; Wang et al., 2020a; Yu et al., 2020). In addition, many studies (Thornton et al., 2010;
Mielke et al., 2011; Phillips et al., 2012; Riedel et al., 2013; Bannan et al., 2015; Faxon et al., 2015;
Wang et al., 2017b; Wang et al., 2017c; Tham et al., 2018; Wang et al., 2018) have also reported
significant amounts of ClNO$_2$ at various continental sites with limited marine influence. For
example, ClNO$_2$ concentrations reached 4 ppbv in the summer of North China Plain (Tham et al.,
2016). These observations imply the importance of other sources for aerosol chloride, such as coal
combustion (Eger et al., 2019), biomass burning (Ahern et al., 2017), waste incineration (Bannan
et al., 2019), and snow-melting agent application (Mielke et al., 2016; McNamara et al., 2020).

In addition to insoluble minerals (e.g., quartz, feldspar, clay and carbonate), mineral dust

aerosols emitted from saline topsoil in arid and semi-arid regions may contain significant amounts
of soluble materials such as chloride and sulfate (Gillette et al., 1992; Abuduwailli et al., 2008;
Zhang et al., 2009; Wang et al., 2012; Jordan et al., 2015; Frie et al., 2017; Gaston et al., 2017;
Tang et al., 2019; Gaston, 2020). As elemental and mineralogical compositions are different for
conventional and saline mineral dust, they would differ significantly in physicochemical properties
and impacts on atmospheric chemistry and climate. For example, hygroscopicity and cloud
condensation nuclei (CCN) activities of saline mineral dust can be much higher than conventional



mineral dust (Pratt et al., 2010; Gaston et al., 2017; Tang et al., 2019; Zhang et al., 2020). Recent
laboratory studies (Mitroo et al., 2019; Royer et al., 2021) found that heterogeneous reactions of
$N_2O_5$ with saline mineral dust originating from western and southwestern USA can be very
effective and produce significant amounts of $ClNO_2$. Large variations in $\gamma(N_2O_5)$ and $\varphi(ClNO_2)$
were reported (Mitroo et al., 2019; Royer et al., 2021), depending on RH as well as chemical and
mineralogical contents of saline mineral dust samples.

A very recent study (Wu et al., 2020) showed that $N_2O_5$ uptake onto saline mineral dust

contributed significantly to particulate nitrate formation during a dust storm event in Shanghai,
China. One may further expect that it may have a profound effect on $ClNO_2$, especially considering
that vast areas in China are heavily affected by both mineral dust and $NO_x$. Nevertheless,
heterogeneous formation of $ClNO_2$ from $N_2O_5$ uptake onto saline mineral dust in other regions
rather than USA has not been explored. In order to provide key parameters required to assess the
potential of saline mineral dust as a $ClNO_2$ source in China, we conducted a series of laboratory
experiments to investigate $ClNO_2$ formation in heterogeneous reaction of $N_2O_5$ with several saline
mineral dust samples collected from different regions in China. In addition to difference in source
regions, saline mineral dust samples examined in our work have substantial variations in
composition and mineralogy, enabling us to examine the effects of particle composition and water
content on $ClNO_2$ production. In order to better understand variations of $ClNO_2$ yields with RH
and samples, we experimentally measured mass hygroscopic growth factors of the eight samples
examined, while previous studies (Mitroo et al., 2019; Royer et al., 2021) used the thermodynamic
model ISORROPIA-II (Fountoukis and Nenes, 2007) to predict particulate water contents. Based
on our laboratory results, we further use a 3-D chemical transport model (GEOS-Chem) to assess





the impacts of $ClNO_2$ produced from $N_2O_5$ uptake onto mineral dust on $ClNO_2$ and $O_3$ in China
during a major dust event which occurred in May 2017.
**2 Methodology**
**2.1 Characterization of saline mineral dust samples**
Eight saline mineral dust samples, originating from five different provinces in northern China
(including Ningxia, Xinjiang, Shandong, Inner Mongolia and Shaanxi), were examined in this
work, and full information of these samples can be found elsewhere (Tang et al., 2019). Table 1
summarizes key information of these samples. According to their chloride contents, the eight
samples were classified into three categories, including two high chloride samples (H1 and H2),
four medium chloride samples (M1, M2, M3 and M4) and two low chloride samples (L1 and L2).
Our previous work (Tang et al., 2019; Zhang et al., 2020) measured mass hygroscopic growth
factors of the eight samples at 0-90% RH with a RH resolution of 10%, using a vapor sorption
analyzer (Gu et al., 2017). As the highest RH at which heterogeneous reaction of $N_2O_5$ with saline
mineral dust was conducted in our work was ~75%, we further measured mass growth factors of
the eight samples at (75±2)% RH, and the results are also included in Table 1.

**Table 1.** Overview of mass fractions of major soluble ions and mass ratios of particulate water at
(75±2)% RH to dry particles for the eight saline mineral dust samples examined in this work. Mass
fractions of major soluble ions were reported previously (Tang et al., 2019), and particulate water
contents at (75±2)% RH were measured by the present work.

| category | sample [a] | sample [b] | $Na^+$ | $Cl^-$ | $SO_4^{2-}$ | $H_2O$ (75%) |
|---|---|---|---|---|---|---|
| High $Cl^-$ | H1 | NX | 0.3537 | 0.3870 | 0.0958 | 1.3093 |
|  | H2 | XJ-5 | 0.2407 | 0.2145 | 0.0973 | 1.7066 |





| | | | | | | |
|---|---|---|---|---|---|---|
| Medium Cl⁻ | M1 | SD | 0.0265 | 0.0508 | 0.0754 | 0.3911 |
| | M2 | XJ-4 | 0.0326 | 0.0341 | 0.0071 | 0.0428 |
| | M3 | IM-2 | 0.0471 | 0.0229 | 0.1413 | 0.2106 |
| | M4 | IM-3 | 0.1343 | 0.0095 | 0.3424 | 0.0174 |
| Low Cl⁻ | L1 | XJ-3 | 0.0239 | 0.0093 | 0.0497 | 0.0475 |
| | L2 | SX | 0.0003 | n.d. | n.d. | 0.0126 |

[a]: sample names used in the present work; [b]:corresponding sample names used in our previous
work (Tang et al., 2019).

**2.2 Experimental apparatus**

Figure 1 shows the experimental apparatus used to study heterogeneous interactions of $N_2O_5$
with saline mineral dust. It mainly consists of three parts: 1) $N_2O_5$ generation, 2) gas-particle
interaction, and 3) detection of $N_2O_5$ and $ClNO_2$.

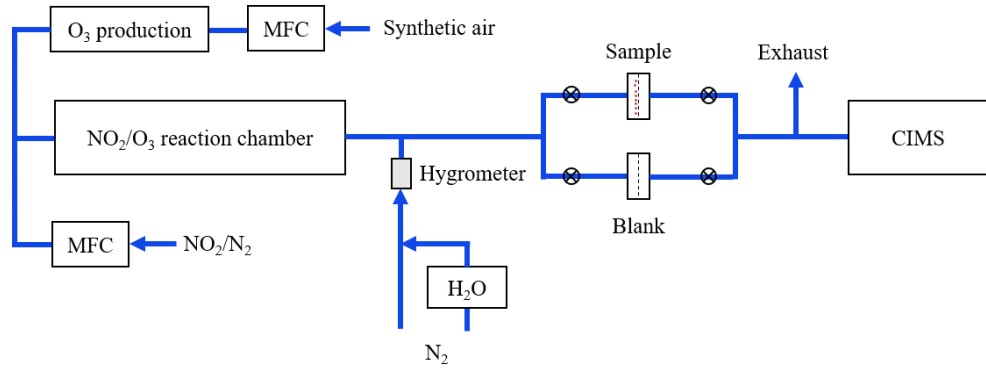


**Figure 1.** Schematic diagram of the experimental apparatus.

**2.2.1 $N_2O_5$ generation**

In our work, $N_2O_5$ was generated via oxidation of $NO_2$ by $O_3$. As shown in Figure 1, a
synthetic air flow (30 mL/min) was passed over a Hg lamp to produce $O_3$ via $O_2$ photolysis at
184.95 nm. The photolysis module was stabilized at $35\pm0.2$ ℃ using a Peltier cooler controlled by
a Proportion Integration Differentiation (PID) algorithm, in order to give stable $O_3$ output. The





O$_3$/air flow was then mixed with a NO$_2$ flow (80 mL/min, 10 ppmv in synthetic air) in a
temperature-stabilized PFA reactor with a residence time of ~70 s to produce N$_2$O$_5$. After exiting
the PFA reactor, the flow (110 mL/min) was then diluted with a humidified nitrogen flow (2500
mL/min), and RH of the humidified nitrogen flow was monitored using a hygrometer. The final
flow had a total flow rate of 2610 mL/min, and initial N$_2$O$_5$ concentrations were in the range of
0.4-1.0 ppbv.
**2.2.2 Heterogeneous interactions**
As shown in Figure 1, the mixed flow (2610 mL/min) could be directed through a blank PTFE
membrane filter (47 mm, Whatman, USA) housed in a PFA filter holder, and in this case initial
N$_2$O$_5$ and ClNO$_2$ concentrations were measured. Alternatively, the flow could also be passed
through a PTFE filter loaded with saline mineral dust, and thus N$_2$O$_5$ and ClNO$_2$ concentrations
after heterogeneous reaction with saline mineral dust loaded on the filter were measured. During
our experiments, the flow could be switched back to pass through the blank filter in order to check
whether the initial N$_2$O$_5$ and ClNO$_2$ concentrations were stable.
Saline mineral dust particles were loaded onto PTFE filters using the method described in our
previous study (Li et al., 2020). PTFE filters were weighted before and after being loaded with
particles, in order to determine the mass of particles loaded onto these filters. In our work, the mass
of particles on filters were in range of 0.6-7.3 mg.
**2.2.3 Detection of N$_2$O$_5$ and ClNO$_2$**
After exiting one of the two filters, a flow of 2200 mL/min was sampled into a time-of-flight
chemical ionization mass spectrometry (TOF-CIMS) to measure N$_2$O$_5$ and ClNO$_2$ concentrations,
and the remaining flow (~400 mL/min) went into the exhaust. The CIMS instrument has been
detailed previously (Kercher et al., 2009; Wang et al., 2016). In brief, N$_2$O$_5$ and ClNO$_2$ were





detected as $I(N_2O_5)^-$ and $I(ClNO_2)^-$ clusters at 235 and 208 m/z (R2a, R2b) using $I^-$ as the reagent
ion, and a soft X-ray device (Hamamatsu, Soft X-Ray 120$^o$) was employed to generate $I^-$ from
$CH_3I/N_2$. CIMS was calibrated before and after our experiments which lasted for ~1 month, and
further details on calibration can be found in the Appendix.
$$N_2O_5 + I^- \rightarrow I(N_2O_5)^- \qquad (R2a)$$
$$ClNO_2 + I^- \rightarrow I(ClNO_2)^- \qquad (R2b)$$
**2.3 Model description**
We use GEOS-Chem (version 12.9.3) to quantify the effects of $ClNO_2$ formation due to
heterogeneous reaction of $N_2O_5$ with saline dust in China. The model, which includes a detailed
representation of coupled ozone-$NO_x$-VOCs-aerosol-halogen chemistry (Wang et al., 2021), is
driven by MERRA2 (the Modern-Era Retrospective Analysis for Research and Applications,
Version 2) assimilated meteorological fields from the NASA Global Modeling and Assimilation
Office (GMAO) with native horizontal resolution of 0.25$^o$×0.3125$^o$ and 72 vertical levels from the
surface to the mesosphere. Our simulation was conducted over East Asia (60°-150°E, 10°S-55°N)
at the native resolution with dynamical boundary conditions from a 4°×5° global simulation.
Anthropogenic emissions in China are based on the Multiresolution Emission Inventory for China
(MEIC) (Zheng et al., 2018) and an inventory of HCl and fine particulate $Cl^-$ in China (Fu et al.,
2018). Natural dust emissions are calculated based on Ridley et al. Ridley et al. (2013). A more
detailed description of the model and emissions can be found elsewhere (Wang et al., 2020b).
For $N_2O_5$ uptake onto aqueous aerosols, the parameterization in our previous study (Wang et
al., 2020b) for $\gamma(N_2O_5)$ and $\varphi(ClNO_2)$, which are based on a detail evaluation of different model
parameterizations by previous work (McDuffie et al., 2018a; McDuffie et al., 2018b), is used in
this study. For $N_2O_5$ uptake on dust aerosol, $\gamma(N_2O_5)$ is always assumed to be 0.02, as





recommended previously (Crowley et al., 2010; Tang et al., 2017), and $\varphi(ClNO_2)$ is assumed to
be 0 in the standard case, i.e., no $ClNO_2$ is produced in heterogeneous reaction of $N_2O_5$ with
mineral dust.

**3 Results and discussion**

Figure 2a shows changes in $N_2O_5$ and $ClNO_2$ concentrations during an experiment in which

heterogeneous reaction of $N_2O_5$ with sample H1 at 37% RH was studied. As shown in Figure 2a,
when the mixed flow was passed through the blank filter (0-10 min), $N_2O_5$ concentrations were
measured to be ~350 pptv and $ClNO_2$ was below the detection limit. The mixed flow was then
passed through the particle-loaded filter at ~10 min in order to initiate heterogeneous reaction of
$N_2O_5$ with sample H1, and significant decrease in $N_2O_5$ concentrations (from ~350 to ~150 pptv)
and increase in $ClNO_2$ concentrations (from almost 0 to ~150 pptv) were observed, suggesting that
heterogeneous interaction with sample H1 substantially consumed $N_2O_5$ and generated $ClNO_2$. In
order to check if initial $N_2O_5$ and $ClNO_2$ concentrations were stable, during our experiments the
mixed flow was switched back to pass through the blank filter from time to time (e.g., at around
40, 75 and 105 min for the experiment displayed in Figure 2a). Indeed, initial $N_2O_5$ and $ClNO_2$
concentrations were constant in our experiments, with another two examples shown in Figures 2b
and 2c.

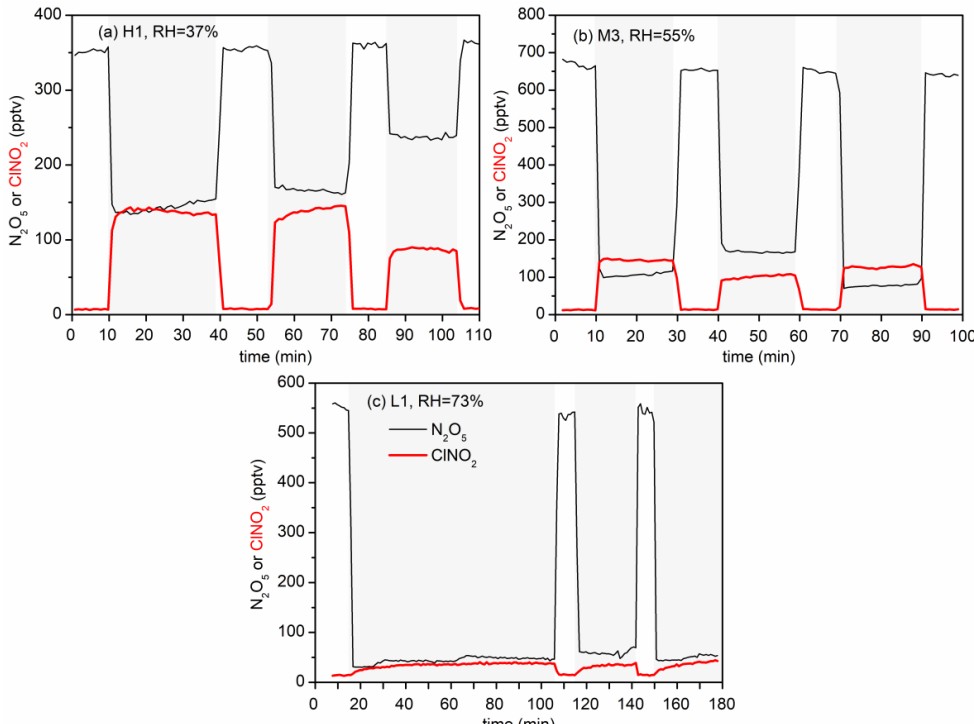


**Figure 2.** Time series for measured $N_2O_5$ and $ClNO_2$ concentrations after the mixed flow was

passed through the blank filter or the particle-loaded filter: a) H1, 37% RH; b) M3, 55% RH; c)

L1, 73% RH. Periods in which the mixed flow was passed through the particle-loaded filter was

shadowed with gray.


Figures 2b and 2c show time series of measured $N_2O_5$ and $ClNO_2$ concentrations in another
two experiments, suggesting that heterogeneous reaction with sample M3 and L1 also led to
substantial removal of $N_2O_5$. However, much less $ClNO_2$ was produced for sample M3 and L1,
when compared to sample H1 (Figure 2a). The decrease in $N_2O_5$ concentrations, $\Delta[N_2O_5]$, and the
increase in $ClNO_2$ concentrations, $\Delta[ClNO_2]$, can be used to calculate $ClNO_2$ yields, $\varphi(ClNO_2)$,
according to Eq. (1).





$$\varphi(\text{ClNO}_2) = \frac{\Delta[\text{ClNO}_2]}{\Delta[\text{N}_2\text{O}_5]} \qquad (1)$$

In this work we measured $\varphi(\text{ClNO}_2)$ for heterogeneous reaction of $\text{N}_2\text{O}_5$ with eight different
saline mineral dust samples at four RH, and each experiment was repeated at least three times. It
should be mentioned that during each experiment the measured $\varphi(\text{ClNO}_2)$ did not vary
significantly with time, and therefore an average value of $\varphi(\text{ClNO}_2)$ was reported for each
experiment. Table 2 summarizes measured $\varphi(\text{ClNO}_2)$ for the eight samples at different RH, and
the results are further discussed in the following sections.

**Table 2.** Measured $\text{ClNO}_2$ yields for heterogeneous uptake of $\text{N}_2\text{O}_5$ onto saline mineral dust
samples at different RH. All the errors given in this work are standard deviations. The uncertainty
of RH was ±2%.

| sample | 18% RH | 36% RH | 56% RH | 75% RH |
|--------|--------|--------|--------|--------|
| H1 | 0.402±0.138 | 0.663±0.039 | 0.774±0.028 | 0.697±0.311 |
| H2 | 0.560±0.046 | 0.474±0.026 | 0.494±0.042 | 0.378±0.069 |
| M1 | 0.271±0.038 | 0.271±0.030 | 0.418±0.053 | 0.543±0.086 |
| M2 | 0.166±0.018 | 0.246±0.041 | 0.316±0.046 | 0.418±0.052 |
| M3 | 0.223±0.061 | 0.251±0.050 | 0.211±0.025 | 0.120±0.050 |
| M4 | 0.179±0.075 | 0.133±0.007 | 0.205±0.021 | 0.181±0.044 |
| L1 | 0.037±0.006 | 0.030±0.015 | 0.045±0.025 | 0.048±0.008 |
| L2 | 0.012±0.003 | 0.005±0.004 | 0.024±0.042 | 0.041±0.039 |


**3.1 ClNO₂ production yields**
Figure 3 shows $\text{ClNO}_2$ yields as a function of RH for the two samples with high chloride
content (H1 and H2), and $\varphi(\text{ClNO}_2)$ were found to be quite high for the two samples. To be more
specific, the mass fraction of chloride was 0.3870 for sample H1, and $\varphi(\text{ClNO}_2)$ were found to
increase from 0.402±0.138 at 18% RH to 0.774±0.028 at 56% RH, and then slightly decreased to
0.697±0.311 when RH was further increased to 75%. For sample H2, the mass fraction of chloride
(0.2145) was lower than sample H1, and $\varphi(ClNO_2)$ showed a small decrease (or remained
relatively constant) when RH was increased from 18% to 56%, ranging from 0.474±0.026 to
0.560±0.046; further increase in RH to 75% resulted in small decrease in $\varphi(ClNO_2)$ to 0.378±0.069.

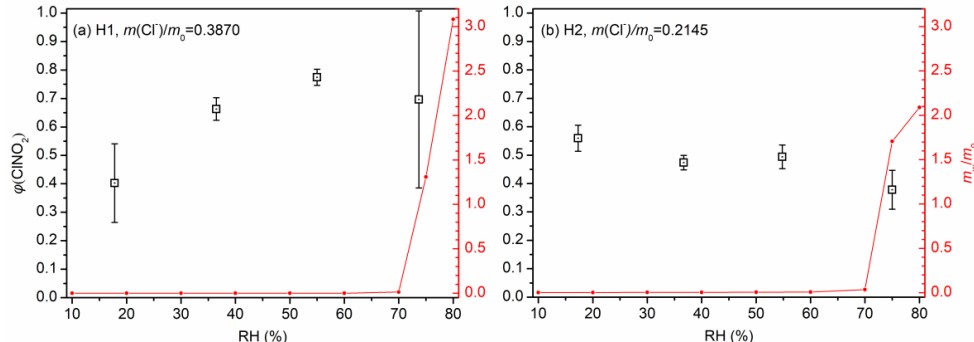


**Figure 3.** Measured $ClNO_2$ yields (black symbol) and normalized mass (normalized to the mass
of dry particles) of particulate water (red line) as a function of RH for (a) H1 and (b) H2. The error
bar represents standard deviation.

$ClNO_2$ yields are shown in Figure 4 as a function of RH for the two low chloride samples (L1

and L2), and their mass fractions of chloride were <0.01. As shown in Figure 4, $\varphi(ClNO_2)$ were
found to be always <0.05 for the two samples, suggesting that heterogeneous production of $ClNO_2$
was very limited, despite substantial removal of $N_2O_5$ due to heterogeneous reaction (with an
example shown in Figure 2c). The low $\varphi(ClNO_2)$ values for sample L1 and L2 could be attributed
to their low chloride contents. In addition, $\varphi(ClNO_2)$ appeared to increase with RH for L1 and L2;
however, since the uncertainties associated with $\varphi(ClNO_2)$ were rather large for these two samples,
the dependence of $\varphi(ClNO_2)$ on RH should be treated in caution.

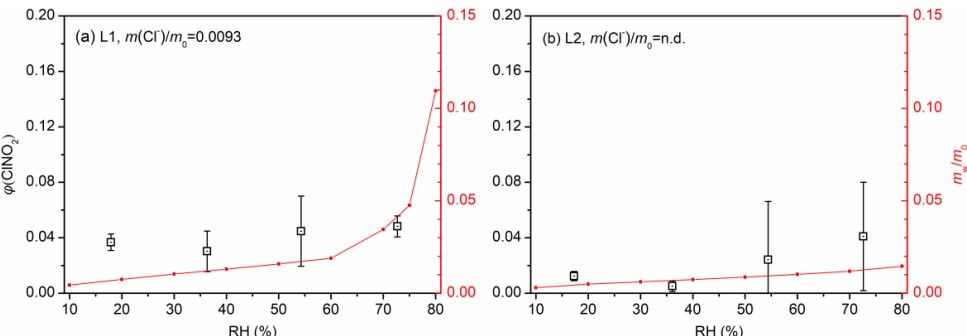


**Figure 4.** Measured $ClNO_2$ yield (black symbol) and normalized mass (normalized to the mass of
dry particles) of particulate water (red line) as a function of RH for (a) L1 and (b) L2 (n. d.: not
detected). The error bar represents standard deviation.

We also investigated $ClNO_2$ production from heterogeneous reaction of $N_2O_5$ with four

samples with medium chloride contents (M1, M2, M3 and M4), and the results are displayed in
Figure 5. Mass fractions of chloride were determined to be 0.0508 for M1, 0.034 for M2, 0.0229
for M3 and 0.0095 for M4, respectively. $ClNO_2$ yields were found to increase significantly with
RH for M1 and M2; more specifically, $\varphi(ClNO_2)$ increased from 0.271±0.038 at 18% RH to
0.543±0.086 at 75% RH for sample M1, and increased from 0.166±0.018 at 18% RH to
0.418±0.0052 at 75% RH for sample M2. As shown in Figure 5, the dependence of $\varphi(ClNO_2)$ on
RH for the other two medium chloride samples (M3 and M4) were rather different from M1 and
M2. For sample M3, $\varphi(ClNO_2)$ first increased from 0.223±0.061 at 18% RH to 0.251±0.050 at 36%
RH, and further increase in RH to 75% caused substantial reduction in $\varphi(ClNO_2)$. At last, no
significant variation of $\varphi(ClNO_2)$ with RH (18-75%) was observed for sample M4.

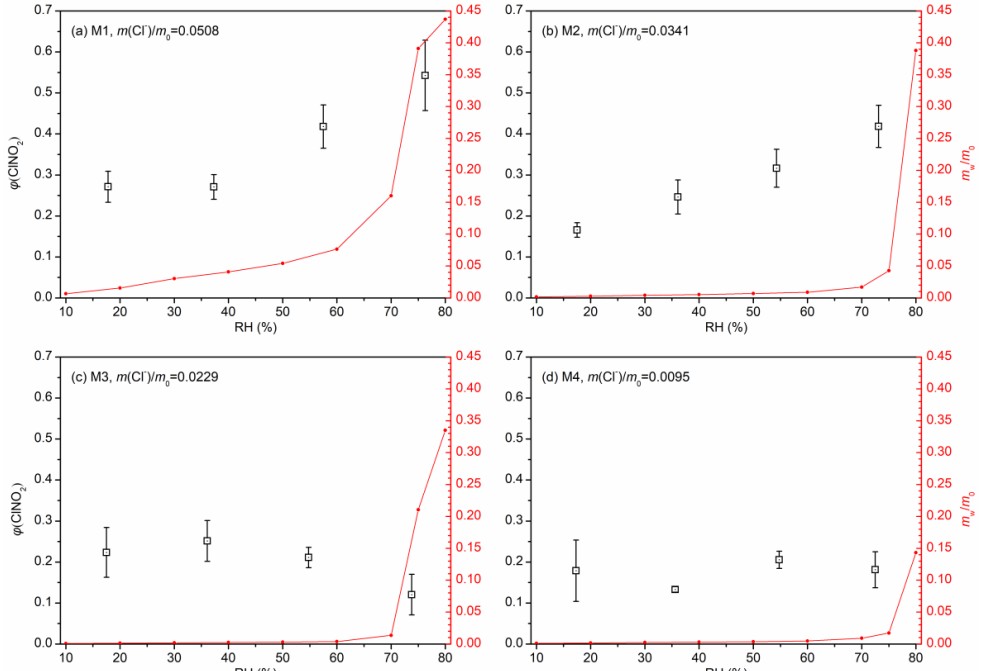

**Figure 5.** Measured $ClNO_2$ yield (black symbol) and normalized mass (normalized to the mass of

dry particles) of particulate water (red line) as a function of RH for (a) M1, (b) M2, (c) M3, and

(d) M4. The error bar represents standard deviation.

**3.2 The effects of RH**

The dependence of $\varphi(ClNO_2)$ on RH for the eight saline mineral samples we examined, as

discussed in Section 3.1, exhibited two interesting features. First, when RH was as low as 18%,

large $\varphi(ClNO_2)$ values (>0.2) were observed for four samples (H1, H2, M1 and M3). As the

deliquescence RH of NaCl is ~75%, one may wonder where aqueous chloride, which is necessary

for heterogeneous formation of $ClNO_2$, came from at 18% RH. As initially suggested by a previous

study (Mitroo et al., 2019), the occurrence of aqueous chloride in saline mineral dust particles at

low RH could be due to the presence of $CaCl_2$ and $MgCl_2$, which were amorphous under dry

conditions and could take up water at very low RH (Guo et al., 2019). Our previous study (Tang





et al., 2019) measured water soluble ions contained by the eight saline mineral dust samples, and
the amounts of water soluble $Ca^{2+}$ and $Mg^{2+}$ in the four samples (H1, H2, M1 and M3) with larger
$\varphi(ClNO_2)$ at 18% RH were significantly larger than those in the other four samples (M2, M4, L1
and L2). This observation further supported our deduction that the presence of $CaCl_2$ and $MgCl_2$
enabled efficient formation of $ClNO_2$ at low RH.

The second interesting feature is that as shown in Figures 3-5, $\varphi(ClNO_2)$ could increase,

decrease or remain relatively constant with increase in RH from 18% to 75%. This feature can be
understood given the complex mechanisms driving heterogeneous uptake of $N_2O_5$ onto saline
mineral dust (Mitroo et al., 2019; Royer et al., 2021): at a given RH, $N_2O_5$ can react with aqueous
water, aqueous chloride and insoluble minerals, and only its reaction with aqueous chloride would
produce $ClNO_2$. The possible effects of RH on $\varphi(ClNO_2)$ are discussed below: 1) as RH increases,
heterogeneous reactivity of $N_2O_5$ towards insoluble minerals can be enhanced, suppressed or
remain largely unchanged (Tang et al., 2012; Tang et al., 2017); 2) increase in RH would lead to
further hygroscopic growth and dilution of aqueous solutions, leading to decrease in $\varphi(ClNO_2)$ in
this aspect; 3) the increase in particulate water with RH would cause more chloride to be dissolved
into aqueous solutions, and in this aspect increase in RH would promote $ClNO_2$ formation. As a
result, it is not surprised to observe different dependence of $\varphi(ClNO_2)$ on RH for different saline
mineral dust samples.
**3.3 Discussion**

Figure 6 shows the dependence of $\varphi(ClNO_2)$ on mass fractions of chloride for the eight

samples we examined at four different RH. These samples showed significant variation in
$\varphi(ClNO_2)$, ranging from <0.1 to >0.7, and $\varphi(ClNO_2)$ were largest for the two high chloride samples
(H1 and H2), followed by median (M1, M2, M3 and M4) and low chloride samples (L1 and L2).



Overall, a positive dependence of $\varphi(ClNO_2)$ on mass fractions of chloride was observed at each
RH. Figure 6 also reveals that the measured $\varphi(ClNO_2)$ were very sensitive to mass fractions of
chloride when the mass fractions of chloride were below 10%. However, as shown in Figure 6,
higher chloride contents did not always mean larger $\varphi(ClNO_2)$, and similar observations were also
reported by previous work (Mitroo et al., 2019; Royer et al., 2021).

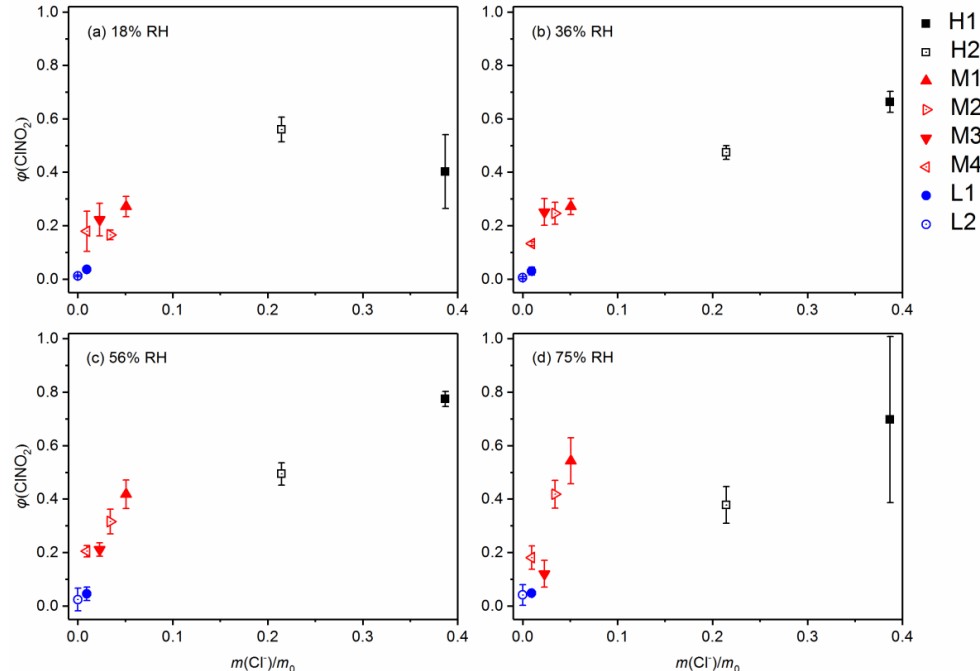

**Figure 6.** Dependence of $ClNO_2$ yields on mass fractions of chloride for the eight saline mineral
dust samples at a given RH: a) 18% RH; b) 36% RH; c) 56% RH; d) 75% RH.

Two parameterizations have been widely used to predict the dependence of $\varphi(ClNO_2)$ on

chemical compositions and water contents of aqueous aerosol particles (Bertram and Thornton,
2009; Yu et al., 2020). Based on laboratory results, Bertram and Thornton (2009) suggested that
$ClNO_2$ yields can be calculated using Eq. (2):





$$\varphi(ClNO_2) = \left(1 + \frac{k(H_2O) \cdot [H_2O_{(aq)}]}{k(Cl^-) \cdot [Cl^-]}\right)^{-1} \quad (2)$$

where $[H_2O_{(aq)}]/[Cl^-]$ is the ratio of molar concentration of $H_2O$ to that of $Cl^-$ in aqueous particles,
and the value of $k(H_2O)/k(Cl^-)$ was suggested to be $1/(483\pm175)$ (Bertram and Thornton, 2009).
Very recently, Yu et al. (2020) examined uptake coefficients of $N_2O_5$ onto ambient aerosol
particles at four different sites in China, and suggested that using a value of $1/(105\pm37)$ for
$k(H_2O)/k(Cl^-)$ would lead to better agreement between measured and predicted uptake coefficients
of $N_2O_5$ (Yu et al., 2020).

The two parameterizations were used in our work to calculate $\varphi(ClNO_2)$ at 75% RH for the

eight saline mineral dust samples we examined. $[H_2O_{(aq)}]/[Cl^-]$ was calculated from the measured
mass growth factors at 75% RH and the mass fractions of chloride, assuming that all the chloride
contained by saline mineral dust samples was dissolved into aqueous solutions at 75% RH. The
comparison between measured and calculated $\varphi(ClNO_2)$ is displayed in Figure 7, suggesting that
both parameterizations significantly overestimated the measured $\varphi(ClNO_2)$ for all the eight saline
mineral dust samples we investigated. A previous study (Mitroo et al., 2019) investigated $\varphi(ClNO_2)$
for heterogeneous uptake of $N_2O_5$ onto saline mineral dust samples collected in southwestern USA,
and similarly they found that the measured $\varphi(ClNO_2)$ were significantly smaller than those
predicted using the parameterization proposed by Bertram and Thornton (2009).

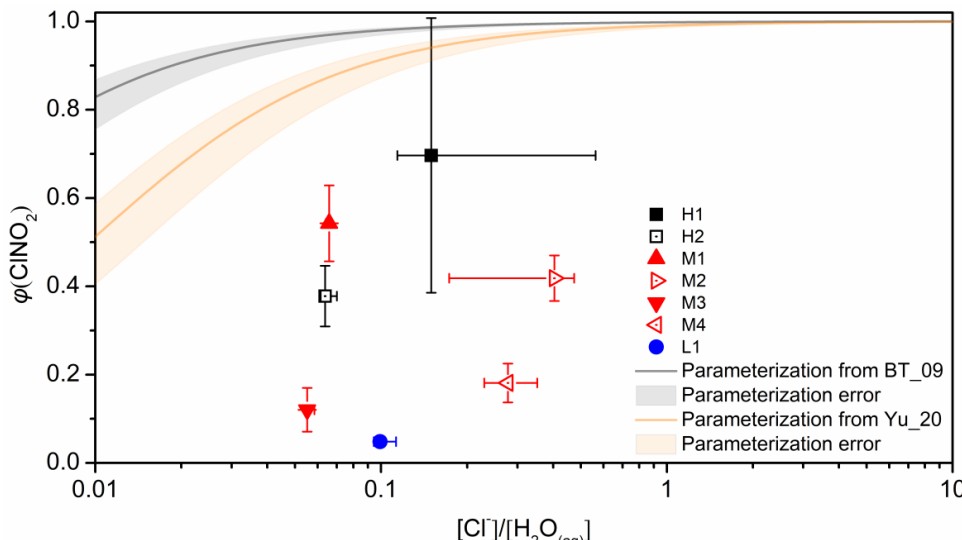


**Figure 7.** Measured and calculated of $\varphi(ClNO_2)$ at 75±2% RH as a function of $[Cl^-]/[H_2O_{(aq)}]$.
Black and orange curves represent $\varphi(ClNO_2)$ calculated using the BT_09 parameterization
(Bertram and Thornton, 2009) and the Yu_20 parameterization (Yu et al., 2020), and the associated
errors are represented by the corresponding shadows.

The observed discrepancies between measured and predicted $\varphi(ClNO_2)$ can be caused by
several reasons. First, even at ~75% RH (the highest RH at which our experiments were conducted),
chloride contained in saline mineral dust may not be fully dissolved, and therefore our calculation
may overestimate $[Cl^-]/[H_2O_{(aq)}]$ and thus also overestimate $\varphi(ClNO_2)$. Second, perhaps more
importantly, saline mineral dust samples contain substantial amounts of insoluble minerals, and
some of these minerals, such as clays, are very reactive towards $N_2O_5$ (Tang et al., 2017); however,
the two parameterizations did not take into account heterogeneous reaction of $N_2O_5$ with insoluble
minerals, and as a result would inevitably overestimate $\varphi(ClNO_2)$. At last, our calculations
assumed internal mixing, but inter- and intra-particle heterogeneity of saline mineral dust particles
could also contribute to the observed gap between measured and calculated $\varphi(ClNO_2)$. For



example, a wintertime field campaign at Ann Arbor (Michigan, USA) (McNamara et al., 2020)
showed that nonhomogeneous chloride distribution across road salt aerosol particles would result
in significant overestimation of $\varphi(ClNO_2)$. The comparison between measured and predicted
$\varphi(ClNO_2)$ suggested that while heterogeneous uptake of $N_2O_5$ onto saline mineral dust could be
an important source of inland $ClNO_2$, underlying mechanisms which affect heterogeneous
production of $ClNO_2$ from saline mineral dust have not been well elucidated.
**4 Atmospheric implications**
We consider $ClNO_2$ formation in heterogeneous uptake of $N_2O_5$ onto dust aerosol in GEOS-
Chem to explore its implications. Since $Cl^-$ concentration in mineral dust is not well known and
currently we are not able to parameterize $\varphi(ClNO_2)$ for mineral dust (as discussed in Section 3.3),
we use a fixed $\varphi(ClNO_2)$ value of 0.1 in our simulation. This value is higher than those determined
in our work for low chloride samples but lower than those for medium chloride samples. We focus
on simulations on 2-7 May 2017, during which a large dust event took place in East Asia. It caused
high concentrations of dust aerosols with maximum hourly concentration higher than 1000 $\mu g/m^3$
over a wide area in China (Zhang et al., 2018), which are also well captured by our simulations
(Figure S1).
Figure 8 shows the weekly mean values of the nighttime maximum surface $ClNO_2$ mixing
ratios and the contribution of heterogeneous reaction of $N_2O_5$ with dust aerosol to $ClNO_2$ over 2-
7 May 2017. The impact of $N_2O_5$ uptake onto dust aerosol is calculated as the difference between
the standard case in which $\varphi(ClNO_2)$ is assumed to be 0 for $N_2O_5$ uptake onto dust aerosol and the
case in which $\varphi(ClNO_2)$ is assumed to be 0.1. Due to large diurnal variations and near-zero mixing
ratios of $ClNO_2$ in the daytime, we use the mean nighttime maximum value for $ClNO_2$, following
previous standard practice (Wang et al., 2019). The largest impact on $ClNO_2$ is found in Central
China, where weekly mean nighttime maximum surface $ClNO_2$ mixing ratios are increased by 85
pptv, due to heavy impact of dust aerosol transported from the north and high $NO_x$ emissions in
this region. Even larger effects (up to 240 pptv increase in $ClNO_2$) can be found on some individual
days, as shown in Figures S2 and S3. These results suggest that $N_2O_5$ uptake onto dust could be
an important source for tropospheric $ClNO_2$ over Central and Northeast China, where $ClNO_2$
formation is conventionally believed to be limited due to relatively low aerosol chloride levels
from sea salts and anthropogenic sources.

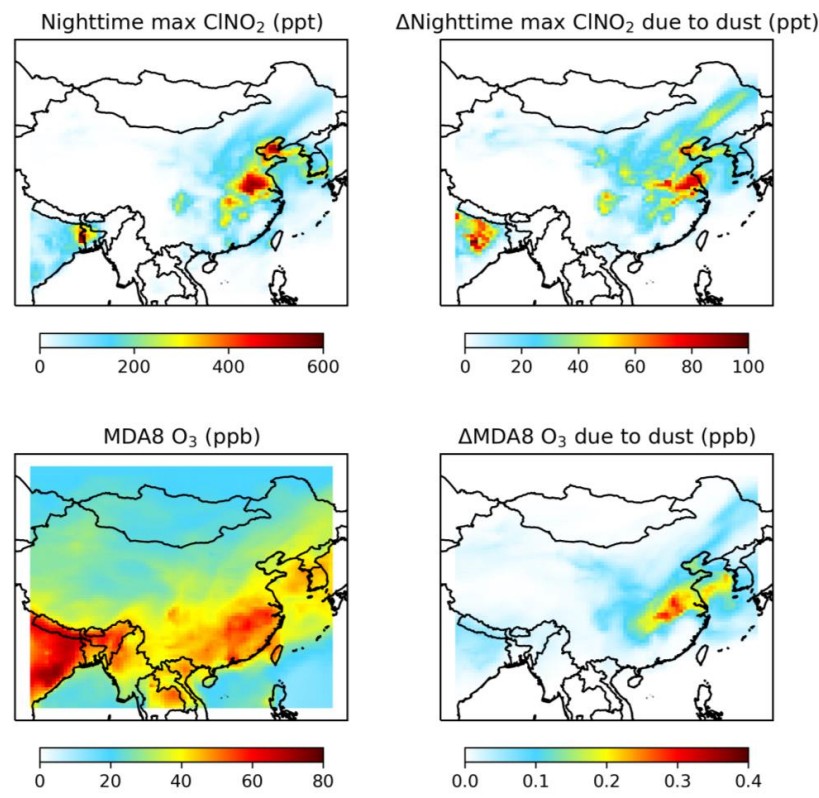


**Figure 8.** Modeled weekly mean mixing ratios of nighttime maximum $ClNO_2$ (upper panels) and
maximum daily 8-h average (MDA8) ozone (bottom panels) in surface air over China during 2-7
May 2017. The left panels show simulated mixing ratios in our standard case in which $\varphi(ClNO_2)$



is assumed to be 0 for $N_2O_5$ uptake onto dust aerosol. The right panels show impacts of $ClNO_2$
formation due to $N_2O_5$ uptake onto dust, calculated as the difference between the standard case
and the case in which $\varphi(ClNO_2)$ is assumed to be 0.1 for $N_2O_5$ uptake onto dust.

Figure 8 also shows the effect of $ClNO_2$ formation due to heterogeneous reaction of $N_2O_5$
with dust aerosol on the daily maximum 8-h average (MDA8) ozone mixing ratios in the surface
air over China during the same period. MDA8 ozone mixing ratios are increased by up to 0.32
ppbv after considering mineral dust as an additional source of $ClNO_2$. Our simulation assumes a
low value of $\varphi(ClNO_2)$ in our measured range (<0.05 to ~0.77), and is conducted in summer when
$ClNO_2$ is more difficult to be accumulated due to short night. We expect that its impacts on $ClNO_2$
and ozone could be larger for dust events in winter and spring.

## 5 Conclusions

It has been widely recognized that nitryl chloride ($ClNO_2$), produced by heterogeneous
reaction of $N_2O_5$ with chloride-containing aerosols, could significantly affect atmospheric
oxidation capacity. However, heterogeneous formation of tropospheric $ClNO_2$ in inland regions in
China has not been well elucidated. In this work, we investigated $ClNO_2$ formation in
heterogeneous reaction of $N_2O_5$ with eight saline mineral dust samples collected from different
regions in China as a function of RH (18-75%). Significant production of $ClNO_2$ was observed for
some of the saline mineral dust samples examined, and $ClNO_2$ yields, $\varphi(ClNO_2)$, were determined
to span from <0.05 to 0.77, depending on chemical compositions of saline mineral dust samples
and RH. In general a positive dependence of $\varphi(ClNO_2)$ on mass fractions of particulate chloride
was observed at each RH, but higher particulate chloride content did not always mean larger
$\varphi(ClNO_2)$. On the other hand, increase in RH could increase, reduce or have no significant impacts





on $\varphi(ClNO_2)$, revealing the complex mechanisms which drive heterogeneous uptake of $N_2O_5$ onto
saline mineral dust.

Two widely-used parameterizations (Bertram and Thornton, 2009; Yu et al., 2020) were used

to estimate $\varphi(ClNO_2)$ at 75% RH for the eight saline mineral dust samples we investigated. Both
parameterizations were found to significantly overestimate the measured $\varphi(ClNO_2)$, and we
suggested that the discrepancies between measured and predicted $\varphi(ClNO_2)$ could be due to
incomplete dissolution of particulate chloride, heterogeneous reaction of $N_2O_5$ with insoluble
minerals, and/or inter- and intra-particle heterogeneity of saline mineral dust particles.

Assuming a $\varphi(ClNO_2)$ value of 0.1 for heterogeneous reaction of $N_2O_5$ with mineral dust, we

use GEOS-Chem to assess the impact of this reaction on tropospheric $ClNO_2$ and $O_3$ in China
during a severe dust event on 2-7 May 2017. It is found that after taking into $ClNO_2$ production
due to $N_2O_5$ uptake onto mineral dust aerosol, weekly mean nighttime maximum $ClNO_2$ mixing
ratios could be increased by up to 85 pptv during this period and the daily maximum 8-h average
$O_3$ mixing ratios were increased by up to 0.32 ppbv.

In summary, our work shows that heterogeneous reaction of $N_2O_5$ with saline mineral dust

can be an important source for tropospheric $ClNO_2$ in inland China. This reaction may also
important for tropospheric $ClNO_2$ production in many other regions over the world, as the
occurrence of saline mineral dust aerosols has been reported in various locations, such as Iran
(Gholampour et al., 2015), United States (Blank et al., 1999; Pratt et al., 2010; Jordan et al., 2015;
Frie et al., 2017), and Argentina (Bucher and Stein, 2016). Currently our limited knowledge
precludes quantitative prediction of heterogeneous $ClNO_2$ production from saline mineral dust,
and further investigation is thus warranted.





**Appendix. $N_2O_5$ and $ClNO_2$ calibration**

To calibrate CIMS measurements of $N_2O_5$, a mixed flow containing $N_2O_5$, which was produced via $O_3$ oxidation of $NO_2$, was sampled into the CIMS instrument, and $N_2O_5$ was quantified using the normalized intensities of $I(N_2O_5)^-$ clusters, $f(N_2O_5)$, defined as the ratio of signal intensity (cps) of $I(N_2O_5)^-$ to that of the total reagent ions, i.e. $I^-$ and $I(H_2O)^-$. $N_2O_5$ concentrations in the mixed flow were quantified using cavity-enhanced absorption spectroscopy (CEAS) (Wang et al., 2017a), with a detection limit of 2.7 pptv in 5 s and an uncertainty of ~25%. RH of the mixed flow was varied during the calibration in order to determine the CIMS sensistivity for $N_2O_5$ at different RH, and the results are displayed in Figure A1. The sensitivity for $N_2O_5$ first increased with RH, reaching the maximum value at ~40% RH, and then decreased with further increase in RH.

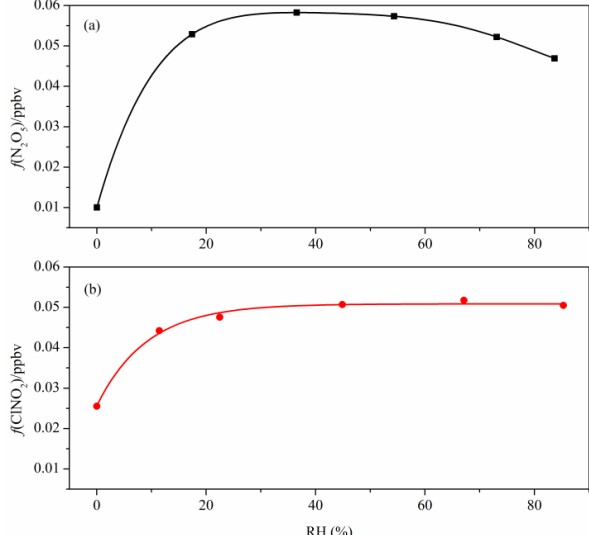

**Figure A1.** CIMS sensitivities as a function of RH for (a) $N_2O_5$ and (b) $ClNO_2$.



To calibrate CIMS measurements of $ClNO_2$, a nitrogen flow (6 mL/min) containing 10 ppmv
$Cl_2$ was passed over a slurry containing $NaNO_2$ and NaCl to produce $ClNO_2$ (Thaler et al., 2011),
and NaCl was included in the slurry in order to minimize the formation of $NO_2$ as a byproduct.
The mixed flow containing $ClNO_2$ was then conditioned to a given RH and sampled into the CIMS
instrument; similar to $N_2O_5$, $ClNO_2$ was quantified using the normalized intensities of $I(ClNO_2)^-$
clusters, $f(ClNO_2)$, defined as the ratio of signal intensity (cps) of $I(ClNO_2)^-$ to that of the total
reagent ions. To quantify $ClNO_2$, the mixed flow was delivered directly into a cavity attenuated
phase shift spectroscopy instrument (CAPS, Model N500, Teledyne API) to measure background
$NO_2$ concentrations; after that, the mixed flow was delivered through a thermal dissociation model
at 365 °C to fully decompose $ClNO_2$ to $NO_2$, and the total $NO_2$ concentrations were then
determined using CAPS. The differences in the measured $NO_2$ concentrations with and without
thermal dissociation was equal to $ClNO_2$ concentrations. The CAPS instrument had a detection
limit of 0.2 ppbv in 1 min for $NO_2$ and an uncertainty of ~10%. As shown in Figure A1, the
sensitivity for $ClNO_2$ increased with RH up to 40%, and showed little variation with further
increase in RH.
The detection limits of CIMS were 2 pptv for $N_2O_5$ and 3 pptv for $ClNO_2$, calculated as four
times of standard deviations ($4\sigma$) when measuring blank samples with 1 min average, and the
accuracy was estimated to be ~25%.



**Data availability**

Data used in this paper can be found in the main text or supplement. GEOS-Chem model is available at GEOS-Chem repository (http://www.geos-chem.org).

**Competing interests**

The authors declare that they have no conflict of interest.

**Author contribution**

**Haichao Wang:** investigation, formal analysis, writing-original draft, writing – review & editing;

**Chao Peng:** investigation, formal analysis, writing-original draft, writing – review & editing;

**Xuan Wang:** investigation, formal analysis, writing-original draft, writing – review & editing;

**Shengrong Lou:** resources; **Keding Lu:** resources, supervision; **Guicheng Gan:** investigation;

**Xiaohong Jia:** investigation; **Xiaorui Chen:** investigation; **Jun Chen:** supervision; **Hongli Wang:** resources; **Shaojia Fan:** resources; **Xinming Wang:** resources; **Mingjin Tang:** conceptualization, formal analysis, resources, supervision, writing-original draft, writing-review & editing.

**Financial support**

This work was funded by National Natural Science Foundation of China (41907185, 91744204 and 42022050), Ministry of Science and Technology of China (2018YFC0213901), Guangdong Basic and Applied Basic Research Fund Committee (2020B1515130003), National State Environmental Protection Key Laboratory of Formation and Prevention of Urban Air Pollution Complex (CX2020080094 and CX2020080578), Guangdong Foundation for Program of Science and Technology Research (2019B121205006 and 2020B1212060053), Guangdong Science and Technology Department (2017GC010501) and CAS Pioneer Hundred Talents program.





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
