# Peer review of "N2O5 uptake onto saline mineral dust: a potential missing source of tropospheric CINO2 in inland China"

_Atmospheric Chemistry and Physics, 2021_

## Author Comment (AC1)

Comments by referees are in blue.
Our replies are in black.
Changes to the manuscript are highlighted in red both here and in the revised manuscript.

**Reply to referee #1**
This paper investigated the heterogeneous $N_2O_5$ uptake and $ClNO_2$ production on the saline mineral dust through laboratory experiments, and evaluated the impacts of this heterogeneous process on tropospheric $ClNO_2$ using a 3-D model. The results showed substantial formation of $ClNO_2$ from the heterogeneous process on different saline mineral dust samples, and the $ClNO_2$ yield varied with the mass fraction of particulate chloride and RH. The model simulation also showed significant impacts of this heterogeneous process on $ClNO_2$ production and even $O_3$ formation during a severe dust event in China. This study provides valuable information on the heterogeneous process of $N_2O_5$ and $ClNO_2$ on saline mineral dust particles, the information of which has been very limited. The results will be useful to better understand the impacts of this heterogeneous process in different environments, and also will be helpful to improve the air quality model performance. Overall, the manuscript is well written, and thus I suggest that the manuscript can be published after addressing the following comments.

**Reply:** We would like to thank ref#1 for reviewing our manuscript and recommending it for publication after minor revision. We have carefully addressed all the comments and revised our manuscript accordingly, as detailed below.

Line 152-153, please clarify what does the 'initial $N_2O_5$ concentrations' mean. Does the author mean the $N_2O_5$ generated from the reaction chamber or before passing the sample filters?

**Reply:** It is a good point. In response to this comment, we have deleted the sentence (in Section 2.2.1) describing initial $N_2O_5$ concentrations, and provided this information in Section 2.2.2 (page 9) we have revised this sentence for better clarity: "As shown in Figure 1, the mixed flow (2610 mL/min) could be directed through a blank PTFE membrane filter (47 mm, Whatman, USA) housed in a PFA filter holder, and in this case initial $N_2O_5$ and $ClNO_2$ concentrations were measured; in our experiments, initial $N_2O_5$ concentrations were in the range of 0.4-1.0 ppbv." After revision, it is clear that initial $N_2O_5$ concentrations are those in the mixed flow immediately before it is passed through the sample filter.

Line 162, although the dust particle loading method has been introduced in previous studies, a brief description will be useful and should be included here.

**Reply:** In the revised manuscript (page 9) we have added one sentence to further describe how we prepared our filter samples: "Saline mineral dust particles were loaded onto PTFE filters using the method described in our previous study (Li et al., 2020; Jia et al., 2021). In brief, 10 mL particle/ethanol mixture was transferred onto a PTFE filter, and after ethanol was evaporated a relatively uniform particle film, as revealed by visual inspection, was formed on the filter."

Line 193-196 and Line 373-375. In addition to the uptake and yield on dust particles, the parameters used for non-dust particles also should be explicit. Some information needs to be briefly provided in the main text or supplementary.

**Reply:** As suggested, we have made the following change in the revised manuscript (page 11): "…is used in this study, and more details can be found in the supplement." We have also updated the supplement accordingly to describe the parameterization we used.

Line 201, the detection limit for these species should be provided in the experimental section.

**Reply:** Detection limits can be found in the Appendix. As suggested by the referee, in the revised manuscript we have moved such information to Section 2.2 (page 10): "The detection

limits were 2 pptv for $N_2O_5$ and 3 pptv for $ClNO_2$, calculated as four times of standard deviations (4 σ) when measuring blank samples with 1 min average, and the accuracy was estimated to be ~25%."

Table 2. Considering the errors given by the standard deviation, the author should avoid using excessive significant digits. This also needs to be checked thoroughly for the whole manuscript.

**Reply:** We have thought carefully about this comment. The referee is absolutely right, and it may be more proper to use 0.01 for the significant digits. However, as shown in Table 2, $ClNO_2$ yields are <0.01 in some cases, and therefore we would like to use 0.001 for the significant digits.

Figure 3. Please clarify the meaning of particulate water, and definition of $m_w/m_0$.

**Reply:** In the revised manuscript (page 14) we have modified the caption of Figure 3 to clarify the meaning of particulate water and define $m_w/m_0$: "Measured $ClNO_2$ yields (black symbol) and $m_w/m_0$ (red line) as a function of RH for (a) H1 and (b) H2. The error bar represents standard deviation, and $m_w/m_0$ represents normalized mass of particulate water (normalized to the mass of dry particles), which was measured as the relative increase in particle mass at a given RH compared to <1% RH." In addition, we have also modified the captions for Figure 4 (page 15) and 5 (page 16) accordingly.

Line 285-290. It's interesting to see that the Ca and Mg amount may affect the $ClNO_2$ yield. Can any figures or plots better depict the dependence of $ClNO_2$ yields on Ca and Mg concentration or fraction in the saline mineral dust samples?

**Reply:** As suggested, to better illustrate the effects of $Ca^{2+}$, in the revised manuscript (page 17) we have made the following modification: "…and as shown in Figure S1, the amounts of water soluble $Ca^{2+}$ in the four samples (H1, H2, M1 and M3) with larger $\varphi(ClNO_2)$ at 18% RH were significantly larger than those in the other four samples (M2, M4, L1 and L2)." In addition, a new figure (Figure S1) has been added into the SI accordingly.

$Mg^{2+}$ may not play an important role in $ClNO_2$ production lower RH, as in our saline mineral dust samples $Mg^{2+}$ usually appears together with $SO_4^{2-}$ (instead of $Cl^-$).

Line 331-332, as the author stated later, the assumption that all chloride is soluble may lead to overestimated $ClNO_2$ yield. What would be a more reasonable assumption here, any semi-quantitative information on the water-soluble Cl fraction/content can be inferred? Is there previous data that can be used to compare the $[Cl]/[H_2O]$ ratio on the dust samples with the normal ambient particles? I think this will be very useful for further modeling simulation works.

**Reply:** This is a good point. As $[Cl^-]$ in aqueous solutions change dynamically with RH, currently we cannot directly measured $[Cl^-]/[H_2O]$. We can use aerosol thermodynamic models to calculate $[Cl^-]/[H_2O]$; however, our previous work (Zhang et al., 2019) found that ISORROPIA-II failed to predict aerosol liquid water contents for some of the saline mineral dust samples.

On the other hand, at 75% RH most of (if not all) chloride should be dissolved into aqueous phase, as significant water uptake was observed at 75% RH. In the revised manuscript (page 20) we have added one sentence to discuss this issue: "First, even at ~75% RH (the highest RH at which our experiments were conducted), chloride contained in saline mineral dust may not be fully dissolved, and therefore our calculation may overestimate $[Cl^-]/[H_2O(aq)]$ and thus also overestimate $\varphi(ClNO_2)$. This effect should not be large as significant water uptake was observed at ~75% RH for saline mineral dust sample we examined (Figures 3-5)."

Line 365, can the author explain more the rationale for choosing 0.1 as the fixed ClNO2 yield in the model simulation?

**Reply:** In the revised manuscript (page 21) we have made the following change to further explain the rationale for choosing 0.1 as the $ClNO_2$ yield and to further explain the purpose of our

modeling work: "This value, which is at the low end of our measured range of $\varphi(ClNO_2)$ (<0.05 to ~0.77), is higher than those determined in our work for low chloride samples but lower than those for medium chloride samples. The purpose of our modeling work, is to preliminarily assess whether $N_2O_5$ uptake onto saline dust as a potential source of $ClNO_2$ may have important effects on tropospheric chemistry."

Line 397-399. The 'short' night in summer may still be enough to accumulate $ClNO_2$ with plenty of $NO_x$, $O_3$, and particles. The statements here need further revision and improvement.

**Reply:** This referee is right. What we actually want to express is that $ClNO_2$ may be more important in winter and spring (due to longer nights) than summer (due to shorter nights). In the revised manuscript (page 24) we have modified the sentence to make our statement more proper: "…and is conducted in summer when $ClNO_2$ is more difficult to be accumulated due to short night (compared to winter and spring with long nights)."

---

## Author Comment (AC2)

Comments by referees are in blue.
Our replies are in black.
Changes to the manuscript are highlighted in red both here and in the revised manuscript.

**Reply to referee #2**

Nitryl chloride ($ClNO_2$) is an important precursor of atmospheric chloride radical, which influences the atmospheric oxidation and regulates the fate of air pollutants. This work conducted a comprehensive lab study of $ClNO_2$ formation from $N_2O_5$ uptake on eight kinds of saline mineral dust samples collected from different regions in China. The result shows that the $ClNO_2$ yield largely impacted by the chloride contents in the saline mineral dust, but the relative humidity seems have no consistent rule in influencing the yield, indicating a complicated relationship between RH and the yield. Further simulation by GEOS-CHEM model demonstrates that the heterogeneous uptake of $N_2O_5$ on saline mineral dusts acted as an important source for the atmospheric $ClNO_2$ during the dust event over China. Overall, this topic is interesting and within the scope of ACP, the data analysis is sound and the manuscript is well written. It can be considered to accept after addressing the following several minor comments.

**Reply:** We would like to thank ref#2 for reviewing our manuscript and recommending it for publication after minor revision. We have carefully addressed all the comments and revised our manuscript accordingly, as detailed below.

Line 40, suggest adding a phrase such like "in addition" before the sentence "Assuming a uniform φ(ClNO2)…", to make clear that the subsequent contents have no relationship with the previous sentence "We further found that current parameterizations significantly overestimated φ(ClNO2)…".

**Reply:** We agree with the referee, and the following change has been made in the revised manuscript (page 3): "In addition, assuming a uniform $\varphi(ClNO_2)$ value of 0.10 for $N_2O_5$ uptake onto mineral dust…"

Line 214, in Fig. 2, the RH values are not completely consistent with those listed in Table 2, pleased confirm them.

**Reply:** Although we would like to keep RH very constant, the actual RH fluctuated in different experiments by ±2%, as we stated in the title of Table 2. Considering the uncertainties in RH, RH values in Figure 2 are consistent with those in Table 2.

Line 244, in Fig. 3(b), the value of $m_w$ (wet particle mass) to $m_0$ (dry particle mass) under different humidity should be usually in the range from 1 to more than 1 due to the hygroscopic effect. The current values can be delta$m/m_0$ (the ratio of mass difference to the dry mass). Suggest correcting it if there is any mistake. Same comment is put forward for Fig. 4 and 5.

**Reply:** In work $m_w/m_0$ represents the relative mass of water, equal to the relative mass increase due to hygroscopic growth. In the revised manuscript (page 14) we have modified the caption of Figure 3 to better define $m_w/m_0$: "Measured $ClNO_2$ yields (black symbol) and $m_w/m_0$ (red line) as a function of RH for (a) H1 and (b) H2. The error bar represents standard deviation, and $m_w/m_0$ represents normalized mass of particulate water (normalized to the mass of dry particles), which was measured as the relative increase in particle mass at a given RH compared to <1% RH." In addition, we have also modified the captions for Figure 4 and 5 accordingly.

Line 310, Figure 6 shows that when the mass ratio of Cl to total less than 0.1, the increase in $ClNO_2$ yield with respect to the increasing Cl content seems more significant at high RH condition (56% and 75%), is it possible that high RH promote the dissolution of chloride into the aerosol liquid water?

**Reply:** This is a very good point. In the revised manuscript (page 18) we have added two sentences to mention and discuss this issue: "Furthermore, Figure 6 suggests that when mass fractions of chloride were <10%, the dependence of $\varphi(ClNO_2)$ on Cl contents was stronger at higher RH. This is because increase in RH would promote dissolution of chloride to aqueous water and thus enhance $ClNO_2$ formation."

Line 348, here the inconsistent results between measurement and calculation may be due to the overestimated $[Cl^-]/[H_2O(aq)]$, but another possibility is that compounds suppressed the formation of $ClNO_2$ or compete with $Cl^-$ to react with $NO_2^+$. I encourage the authors to do some discussion.

**Reply:** In the revised manuscript (page 20) we have made the following changes to explain further why the presence of insoluble minerals could suppress $ClNO_2$ formation: "Second, perhaps more importantly, saline mineral dust samples contain substantial amounts of insoluble minerals, and some of these minerals, such as clays, are very reactive towards $N_2O_5$ (Tang et al., 2017), and only nitrate but no $ClNO_2$ was formed (Seisel et al., 2005; Karagulian et al., 2006; Tang et al., 2012)."

Line 355-357, the sentence is not very clear. Nonhomogeneous chloride distribution across road salt aerosol particles during the field observation resulted in higher $ClNO_2$ yield than the theory prediction, right?

**Reply:** The observed yields are lower than the predicted values. In the revised manuscript (page 21) we have made the following modification for better clarity: "…showed that due to nonhomogeneous chloride distribution across road salt aerosol particles, observed $\varphi(ClNO_2)$ were significantly smaller than predicted values."